
# Biogenic, urban, and wildfire influences on the molecular composition of dissolved organic compounds in cloud water

Ryan D. Cook[1†], Ying-Hsuan Lin[1,2,†,‡], Zhuoyu Peng[1Δ], Eric Boone[1,¥], Rosalie K. Chu[3],
James E. Dukett[4], Matthew J. Gunsch[1], Wuliang Zhang[1£], Nikola Tolic[3], Alexander
Laskin[3◊], Kerri A. Pratt[1,5]

[1] Department of Chemistry, University of Michigan, Ann Arbor, MI, USA
[2] Michigan Society of Fellows, University of Michigan, Ann Arbor, MI, USA
[3] Environmental Molecular Sciences Laboratory, Pacific Northwest National Laboratory, Richland, WA
USA
[4] Adirondack Lake Survey Corporation, Ray Brook, NY, USA
[5] Department of Earth & Environmental Sciences, University of Michigan, Ann Arbor, MI, USA
[†] Authors contributed equally to this work.
[‡] Now at Department of Environmental Sciences, University of California, Riverside, Riverside, CA,
USA
[Δ] Now at Department of Chemistry, University of Washington, Seattle, WA, USA
[¥] Now at College of Veterinary Medicine, Purdue University, West Lafayette, IN, USA
[£] Now at Department of Chemistry, Northwestern University, Evanston, IL, USA
[◊] Now at Department of Chemistry, Purdue University, West Lafayette, IN, USA

*Correspondence to:* Kerri A. Pratt (prattka@umich.edu)

**Abstract.** Organic aerosol formation and transformation occurs within aqueous aerosol and cloud droplets, yet little is known about the composition of high molecular weight organic compounds in cloud water. Cloud water samples collected at Whiteface Mountain, New York during August-September 2014 were analyzed by ultrahigh-resolution mass spectrometry to investigate the molecular composition of dissolved organic carbon, with focus on sulfur- and nitrogen-containing compounds. Organic molecular composition was evaluated in the context of cloud water inorganic ion concentrations, pH, and total organic carbon concentrations to gain insights into the sources and aqueous phase processes of the observed high molecular weight organic compounds. The molecular composition of the cloud water depended on the influencing sources (biogenic, urban, wildfire) and showed evidence of aqueous-phase processes. Cloud water acidity was correlated with the average oxygen:carbon ratio of the organic constituents, suggesting the influence of aqueous acid-catalyzed oxidation processes. Organosulfate compounds, with both biogenic and anthropogenic volatile organic compound precursors, were detected for cloud water samples influenced by air masses that had traveled over forested and populated areas. Oxidation products of long-chain ($C_{10-12}$) alkane precursors were detected during urban influence. Influence of Canadian wildfires resulted in increased numbers of identified sulfur-containing compounds and oligomeric species, including those formed through aqueous-phase reactions involving methylglyoxal. Light absorbing aqueous-phase products of syringol and guaiacol oxidation were observed in the wildfire-influenced samples, and dinitroaromatic compounds were observed in all cloud water samples (wildfire, biogenic/urban, and urban-influenced).



# 1 Introduction

Aqueous reactions have been suggested as an important source of high molecular weight organic matter in the atmosphere (Ervens et al., 2015; Herrmann et al., 2015). Similar to sulfate formation upon dissolution and oxidation of $SO_2$ in cloud droplets (Mohnen and Kadlecek, 1989), water-soluble gases

can also be incorporated into cloud droplets and undergo aqueous reactions (Blando and Turpin, 2000), depending on the solubility of the dissolved organic gases (Ervens et al., 2011). Modeling has shown that including cloud processing reactions improves prediction of secondary organic aerosol (SOA) mass concentrations (Carlton et al., 2008). However, there is substantial inconsistency between the chemical composition of laboratory generated and ambient organic aerosol, in part likely due to aqueous processing

(Chen et al., 2015). Potential products of aqueous processing include carboxylic acids, esters, organosulfur compounds, polyols, amines, amino acids, and highly oxygenated oligomeric species (Blando and Turpin, 2000; Ervens et al., 2011). However, only limited ambient evidence of cloud processing involving high molecular weight organic compounds has been reported (Boone et al., 2015; Feng and Möller, 2004; Lee et al., 2011, 2012; Pratt et al., 2013; Sagona et al., 2014; van Pinxteren et al.,

2016; Zhao et al., 2013). A recent review found that more than 50% of organic matter in fog and cloud water remains unspeciated (Herckes et al., 2013). Therefore, examination of high molecular weight organic compounds present in cloud water is crucial to improve our current understanding of the complex multiphase chemical processes leading to SOA formation and transformation.

Laboratory studies have shown that aqueous-phase reactions can lead to the formation of

20 organonitrate and organosulfate compounds (Darer et al., 2011; Minerath et al., 2008; Minerath and Elrod, 2009; Perri et al., 2010). Organosulfates are suggested to be formed from acid-catalyzed reactive uptake of epoxides onto sulfate aerosol (Surratt et al., 2010), as well as in-cloud radical-radical reactions involving sulfate radicals (Nozière et al., 2010; Perri et al., 2010; Schindelka et al., 2013). Isoprene and monoterpene-derived organosulfates have been observed in cloud water over the southeastern United

States (Boone et al., 2015; Pratt et al., 2013). Modeling suggests that organosulfate formation most commonly occurs in aqueous aerosol rather than in cloud droplets (McNeill et al., 2012). Evidence of tertiary organosulfate hydrolysis in cloud water was observed in the southeast US when cloud water composition was compared to below-cloud aqueous aerosol, showing another cloud processing pathway (Boone et al., 2015).

Aqueous-phase processes, particularly in aqueous aerosol, may also produce high molecular weight compounds, such as oligomers (Ervens et al., 2011). Laboratory studies have shown the production of oligomers in bulk aqueous solutions from the photochemical oxidation of glyoxal, methylglyoxal, pyruvic acid, and methacrolein (Ervens et al., 2011). Oligomers have been observed in ambient cloud water, suggesting the influences of these aqueous-phase reactions (Boone et al., 2015; Pratt et al., 2013;

Zhao et al., 2013). However, with few studies reporting the molecular composition of high molecular



weight compounds, it is difficult to assess the potential importance of these aqueous-phase derived compounds and their impacts on SOA composition and wet deposition to ecosystems (Hallquist et al., 2009).

In this work, cloud water samples were collected at Whiteface Mountain, NY during August-September 2014 to investigate the influence of various sources and aqueous-phase reactions on the molecular composition of dissolved organic compounds. An in-depth summary of Whiteface Mountain cloud water studies has been published elsewhere (Schwab et al., 2016a). Whiteface Mountain cloud water is more acidic than rainwater, with higher sulfate and nitrate concentrations (Aleksic et al., 2009), which are attributed to fossil fuel combustion (Dukett et al., 2011). Cloud deposition of total soluble sulfur accounts for 80-90% of total sulfur deposition during June-September (Baumgardner et al., 2003). The cloud water samples were analyzed by electrospray ionization (ESI) coupled to Fourier transform ion cyclotron resonance mass spectrometry (FTICR-MS) in negative ion mode to characterize high molecular weight oxidized organic compounds. Nitrogen-containing, sulfur-containing, and oxygenated organic compounds (CHNO, CHNOS, CHOS, and CHO) were identified in the cloud water and examined in the context of inorganic ion concentrations, pH, and total organic carbon (TOC) concentrations to gain further insights into the aqueous-phase transformations of atmospheric organic aerosol.

## 2 Experimental

### 2.1 Sampling collection and analysis for inorganic ions, pH, and TOC

Cloud water samples were collected during August-September 2014 at Whiteface Mountain (44.3659° N, 73.9026° W; summit observatory 1483 m above mean sea level (AMSL)) in the Adirondack Mountains in New York. An in-depth site description is published elsewhere (Schwab et al., 2016b). Sampling times are provided in Table 1. Cloud water collection and analysis efforts by the Adirondack Lake Survey Corporation (ALSC) closely follow the original guidelines established by the EPA Clean Air Status and Trends Network (CASTnet) program in 1994. An omni-direction passive collector (Aleksic and Dukett, 2010) collected cloud water samples when the following conditions were met: cloud liquid water content > 0.05 g m$^{-3}$ (to indicate the presence of clouds), temperature > 2°C (to avoid freezing of the sampler), wind speed > 2 m s$^{-1}$ (to assist with cloud water collection by the sampler), and no precipitation (to isolate cloud droplets only). A field blank was collected at the summit of Whiteface Mountain by pouring deionized water into a sample container. Routine analyses of inorganic ions ($SO_4^{2-}$, $NO_3^-$, $Cl^-$, $Ca^{2+}$, $Mg^{2+}$, $Na^+$, $K^+$, $NH_4^+$), pH, and TOC are performed by the ALSC, as described elsewhere (Baumgardner et al., 2003). Average August-September values from 2010-2015 are presented in (Table S1), and the data can be accessed at http://www.adirondacklakessurvey.org/.

### 2.2 ESI-FTICR-MS analysis



Cloud water samples and a field blank for mass spectrometric analysis were pre-concentrated using solid phase extraction (Strata™-X polymeric SPE sorbents), according to the method of Zhao et al. (2013) to enhance detection and to reduce potential matrix effects from inorganic ions. TOC concentrations of each cloud water sample after SPE ranged from 15-398 mg L$^{-1}$ (Table S2). It should be noted that lower

molecular weight organic compounds (i.e. most isoprene oxidation products), including organosulfates, are expected to be removed during extraction (Zhao et al., 2013).

A 12-tesla Bruker SolariX FTICR mass spectrometer (Billerica, MA) was used to collect ultrahigh-resolution mass spectra of the cloud water samples. The detailed instrument parameters and analysis procedures have been described previously (Kujawinski and Behn, 2006; Minor et al., 2012;

Tfaily et al., 2015). A standard Bruker electrospray ionization (ESI) source was used to generate negatively charged molecular ions. Samples were first diluted in LC-MS grade acetonitrile to similar TOC concentrations 15 to 45 mg L$^{-1}$ and then introduced through a syringe pump at a flow rate of 3.0 μL min$^{-1}$ through a 200 μm i.d. fused silica transfer tube. Experimental conditions were as follows: needle voltage, +4.5 kV; Q1 set to 100 *m/z*; heated glass coated metal capillary operated at 180ºC, and the ion

accumulation time was adjusted for best spectral quality. These parameters were chosen based on previous experiments that optimized complex organic matter characterization (Tfaily et al., 2011, 2012, 2013). Two hundred individual scans were co-added, and the average resolving power (m/Δm) was >450,000 at 451 Da. Mass spectra were calibrated by two dissolved organic matter homologous series separated by 14 Da (–CH$_2$ groups), and the mass accuracy was calculated to be <1 ppm for singly charged ions ranging

across the mass spectral distribution (*m/z* 100–1000). A solvent blank of acetonitrile was also analyzed. A threshold was set to assign peaks with S/N > 15 from the spectra of all samples. Only sample ion peaks with intensities at least 100 times higher than those in either blank or acetonitrile were retained for further data analysis. Assignments were made with elements limited to C, H, O, N, and S, with S=0-1, N=0-2, and the minimum number of O atoms per detected S or N was set to be 3. Assignments were limited to

mass error of ± 0.5 ppm. All $^{13}$C isotopic peaks were removed. These conservative limits of data processing and assignments resulted in approximately 430-2300 peaks identified per sample (up to ~60% of the total number of detected peaks, after removal of $^{13}$C and blank peaks). A complete list of assigned peaks is provided in Table S3. For individual elemental composition $C_cH_hN_nO_oS_s$, double bond equivalents (DBE) were determined by Eq. (1), where c, h, and n are the number of carbon, hydrogen,

and nitrogen atoms in the molecular formula, respectively.

$$DBE = c-h/2+n/2+1, \hspace{4cm} (1)$$

Sulfur and oxygen are divalent, and nitrogen is trivalent here. Therefore, the DBE calculation does not account for tetravalent and hexavalent S or pentavalent N (Mazzoleni et al., 2010).

### 2.3 Backward air mass trajectory analyses



Sources of the air masses arriving at the sampling site were evaluated using NOAA Hybrid Single Particle Lagrangian Integrated Trajectory Model (HYSPLIT) analysis (Stein et al., 2015). Six day backward air mass trajectories were run from the beginning of the sample collection time at a starting height of 1500 m amsl, based on the Whiteface Mountain Observatory altitude (1483 m). Modeled mixed layer depth for

each trajectory was examined to determine the influence of sources (eg. forests and urban areas) within the atmosphere boundary layer. To investigate whether samples were likely influenced by wildfires, the NOAA Hazard Mapping System Fire and Smoke Product (http://www.ospo.noaa.gov/Products/land/hms.html) was used along with the backward air mass trajectories. The smoke product is divided into categories (light, moderate, heavy smoke) as determined

via satellite images collected using GOES, Advanced Very High Resolution Radiometer (AVHRR), and MODerate resolution Imaging Spectroradiometer (MODIS) images.

## 3. Results and Discussion

Overall, Whiteface Mountain cloud water samples show a correlation between water acidity and the average oxygen-to-carbon atomic ratios (O:C) of the high molecular weight compounds detected by

(-)ESI-FTICR MS (Fig. 1). The average pH of the cloud water samples was $4.6 \pm 0.7$ (95% confidence interval), and the average O:C ratio of the high molecular weight negative ions was $0.505 \pm 0.004$. To investigate the role of precursor sources on the identities of the dissolved organic compounds in the cloud water, samples were grouped based on backward air mass trajectories (Fig. S1) apportioned to: (A) biogenic (August 16-17, 2014), (B) urban (September 2, 2014), and (C) wildfire (August 4-6, 2014)

sources. The average O:C ratios of the grouped biogenic, urban, and wildfire-influenced cloud water samples were $0.460 \pm 0.008$ (95% confidence interval), $0.489 \pm 0.007$, and $0.534 \pm 0.005$, respectively. Average H:C ratios and DBE values corresponding to each of the air masses are listed in Table S4. The following sections discuss the cloud water chemical composition as a function of air mass influence.

### 3.1 Biogenic influence

For the cloud water samples collected on August 16-17, 2014 (A1-A3, Table 1), the air mass had traveled from the Canadian boreal forest before passing over Detroit, MI and Buffalo, NY prior to arrival at Whiteface Mountain (Fig. S1). The measured cloud water pH was 4.79-5.25, less acidic than samples collected under greater urban or biomass burning influence, as well as the Aug.-Sept. 2014 average pH ($4.8 \pm 0.1$, 95% confidence interval) (Table 1). These cloud water samples had the lowest nitrate, sulfate,

chloride, and ammonium concentrations compared to the other samples. Notably, the measured TOC and sulfate concentrations were 0.73-2.16 mg $L^{-1}$ and 3.6-9.7 $\mu$M, respectively, which are below the Aug.-Sept. 2014 averages of $3.1 \pm 0.8$ mg $L^{-1}$ and $32 \pm 9$ $\mu$M (Table 1), as well as the Aug.-Sept. 2010-2015 averages (Table S1).



Negative ion mass spectra for the A1-A3 cloud water samples are shown in Figures 2 and S2 with nitrogen-containing, sulfur-containing, and oxygenated organic compounds (CHNO, CHNOS, CHOS, and CHO) compounds highlighted. Corresponding number fractions of the assigned organic compound types, as well as the distributions of the number of carbon atoms per assigned compound, are shown in Figures 2 and S3, as well as Table S5. For the biogenic-influenced cloud water samples (A1-A3), CHO compounds comprise 51-56%, by number, of the dissolved organic species identified (Table S5). CHO compounds potentially arising from aqueous reactions were identified by comparison to laboratory studies of the aqueous photooxidation of α-pinene oxidation products (*cis*-pinonic acid and the surrogate tricarballylic acid) (Aljawhary et al., 2016) and isoprene oxidation products (methyl vinyl ketone and isoprene SOA) (Nguyen et al., 2012a; Renard et al., 2015) (Table 3). CHOS and CHNOS compounds account for 14% and 7-9%, respectively, of the measured organic compounds in the A1 and A2 samples; lower percentages were observed for sample A3 (Table S5), consistent with the lower sulfate concentration in A3 (Table 1). These biogenic-influenced cloud water samples contained several unique CHOS and CHNOS compounds with relatively low O:C (0.05-0.4) and H:C (0.7-1.1) ratios (Fig. 3 and S4), compared to the sulfur-containing organic compounds in other cloud water samples. These unique compounds all contained high DBE values (9 or greater for $C_{16}$-$C_{25}$ molecules), suggesting the presence of multiple double bonds or ring structures. In the Amazon, Kourtchev et al (2016) previously found a similar subset of these unidentified CHOS compounds in SOA samples, suggesting a biogenic influence. Many organosulfates and nitrooxy-organosulfates, previously reported in laboratory studies, ambient aerosols, and cloud water as derived from monoterpene oxidation (Boone et al., 2015; Hamilton et al., 2013; Iinuma et al., 2007a, 2007b, 2009; Kristensen et al., 2016; Kristensen and Glasius, 2011; Lin et al., 2012; Mazzoleni et al., 2012; Nguyen et al., 2012b; Nozière et al., 2010; O'Brien et al., 2014; Pratt et al., 2013; Stone et al., 2012; Surratt et al., 2008), were detected in the biogenic-influenced cloud water samples (Table 2), also consistent with monoterpene emissions along the air mass trajectories (Guenther et al., 2006). In this study, the required solid-phase extraction step precluded the detection of isoprene-derived organosulfates, which would have been removed prior to analysis. Hydrolysis products (or precursors) of most of the identified organosulfates and organonitrates were also found in the cloud water samples (Table 2), consistent with laboratory studies (Darer et al., 2011; Hu et al., 2011) and the previous field observation of organosulfate hydrolysis in cloud water (Boone et al., 2015).

CHNO compounds contributed 26%, by number in the A1 and A2 samples (Table S5). The A3 sample, characterized by less urban influence (Fig. S1), featured a greater number of CHNO compounds, as well as a higher fraction (38%) (Table S5). Among the CHNO compounds, unique compounds with two N atoms (~30% of total CHNO compounds) were observed in the biogenic-influenced samples (A1-A3); these dinitrogen organic compounds were diverse with O:C 0.05-1.17, H:C 0.5-1.88 (Fig.3 and S4), and DBE 4-17. Several likely dinitroaromatics compounds, including $C_6H_6N_2O_6$ (DBE=5), $C_7H_6N_2O_5$ (DBE=6), $C_7H_6N_2O_6$ (DBE=6), and $C_9H_8N_2O_6$ (DBE=7), were identified in these biogenic-influenced




samples only. The formation of dinitroaromatics has been reported to originate from aqueous-phase radical nitration of mono-nitroaromatics (Kroflič et al., 2015). Since no possible mono-nitroaromatics were observed in the samples, this suggests either a different formation pathway or that complete reaction had occurred. Nitroaromatics are light-absorbing brown carbon compounds, which suggests the potential

climate importance of these compounds following cloud droplet evaporation (Laskin et al., 2015).

### 3.2 Urban influence

The cloud water samples collected on September 2 from 06:00-09:00 EDT (B1) and 18:00-21:00 EDT (B2) were characterized by significant urban influence, with air mass trajectories traveling from the southwest across multiple metropolitan areas, including Buffalo, NY, Detroit, MI, and/or Cleveland, OH

(among others), prior to arrival at Whiteface Mountain (Fig. S1). Elevated sulfate (10.1-69.4 μM) and acidity (pH 4.18-4.86), but similar TOC (1.19-2.11 mg L$^{-1}$), were measured for the cloud water samples, compared to the biogenic-influenced period (Table 1). The sulfate concentrations are representative of Aug.-Sept. 2014 averages of $32 \pm 9$ μM (95% confidence interval), as well as Aug.-Sept. 2010-2015 (Table S1).

The mass spectra of the B1 and B2 cloud water samples are shown in Figures 2 and S2. A total of 2276 oxidized organic compounds were identified in the B1 sample, highlighting the diversity and complexity of the cloud water molecular composition. The CHO compounds were the most prevalent at 44-53%, by number, similar to the other air mass influences (Table S5). CHO compounds with assignments consistent with aqueous-phase reaction products, based on laboratory studies, are noted in

Table 3. Notably, CHNO compounds account for 36-43% of the identified compounds, generally much higher than the number fractions (as well as absolute numbers) observed in the biogenic-influenced cloud water samples (Table S5). This is likely due to the influence of elevated NO$_x$ emissions in urban locations due to fossil fuel combustion. The CHNO compounds in these samples ranged from containing 7 to 32 carbon atoms, with a median of C$_{18}$ (Fig. 2). 24% of the CHNO compounds in the urban-influenced

samples were composed of more than 23 carbon atoms, compared to only 3% and 12% of the CHNO in the biogenic and wildfire-influenced samples, respectively, suggesting the influence of long-chain fossil fuel-derived organic precursors in these urban-influenced samples. Recent cloud water studies by Boone et al. (2015) and Zhao et al. (2013) also observed significant number fractions of CHNO compounds; however, the O:C and H:C ratios of the CHNO compounds observed in these previous studies are not

similar to those measured herein. Based on comparison with below-cloud particles, Boone et al. (2015) suggested aqueous-phase formation of the CHNO compounds, and due to wintertime sampling, Zhao et al. (2013) suggested contribution from residential wood burning.

            CHOS and CHNOS account for 7-10% and 3-4%, by number, respectively, of the identified organic compounds in the urban-influenced samples (Table S5). A fraction of the CHOS compounds,

unique to sample B1, were characterized by low O:C (0.05-0.25) and H:C (0.6-1.2) ratios, similar to



previous cloud water observations by Zhao et al. (2013). Notably, aliphatic organosulfate species derived from photooxidation of long-chain alkane precursors ($C_{10-12}$), including dodecane, cyclodecane, and decalin, recently observed in laboratory and urban ambient aerosol studies (Riva et al., 2016; Tao et al., 2014), were detected in the B1 cloud water sample (Table 2), consistent with the urban air mass influence.

In a recent study by Boone et al. (2015) in the southeastern US, several of these aliphatic organosulfates were observed in below-cloud atmospheric particles, but not in the cloud water. Similar to the northern biogenic-influenced cloud water, many likely monoterpene-derived organosulfates and nitrooxy-organosulfates (Nguyen et al., 2012b; Surratt et al., 2008), as well as corresponding hydrolysis products (or precursors), were observed in the B1 and B2 cloud water samples (Table 2 and 3), due to the time the

air mass spent over the forested areas of the southeast and midwest United States (Fig. S1).

### 3.3 Canadian wildfire influence

Backward air mass trajectories combined with NOAA Hazard Mapping System and Smoke Product data indicate influence from Canadian wildfire smoke during August 4-6, 2014 at Whiteface Mountain (Fig. 4, Fig. S5). Cloud water pH ranged from 4.05-4.54, generally lower than the biogenic and urban-

15 influenced samples and similar to the August-September 2014 average of $4.8 \pm 0.1$. Notably, there appear to be more oligomeric compounds present in the wildfire samples (Fig. 2, Fig. S2), suggesting that the increase in acidity can be linked to the production of oligomeric species. The cloud water inorganic ion concentrations were the highest observed in this study, although sample C3 had lower inorganic ion concentrations than the C1 and C2 samples (Table 1). Elevated $K^+$ concentrations, commonly used as a

20 tracer of biomass burning (Artaxo et al., 1994), were present in the wildfire samples (1.30-5.38 µM), compared to the non-wildfire conditions (biogenic and urban samples: 0.02-0.87 µM) (Table 1). A positive correlation between $K^+$ and TOC concentrations was observed (Fig. S6), with increased TOC mass concentrations (7.86-16.6 mg $L^{-1}$) in wildfire-influenced cloud water, compared to non-wildfire conditions (0.73-2.16 mg $L^{-1}$), as well as the Aug.-Sept. 2010-2015 averages (Table S1). While the

25 contribution of aqueous SOA formation to the measured TOC cannot be determined in this study, Gilardoni et al. (2016) reported production of light-absorbing SOA and an increase in O:C ratio from aqueous-phase processing of biomass burning emissions.

    Previously, Nguyen et al. (2012b) observed the formation of organosulfates following evaporation and re-dissolution of an aqueous solution (pH 2-4) of sulfuric acid mixed with SOA formed from d-

30 limonene ozonolysis. In this study, numerous possible organosulfates corresponding to the formulas observed by Nguyen et al. (2012b) were identified in urban and wildfire-influenced cloud water samples, but not in the biogenic samples (Table 4). Notably, unlike the common organosulfates, likely formed through reactive uptake of epoxides on aqueous aerosol (Surratt et al., 2010) and listed in Table 2, these organosulfates were primarily observed in the B1, C1, and C2 samples, which were characterized by the

35 highest sulfate cloud water concentrations (30.7, 69.4, and 103.2 µM, respectively) and lowest pH (4.05-



4.18), compared to the other cloud water samples ($SO_4^{2-}$ ≤10.1 μM, pH >4.5) (Table 1). Since these organosulfates were observed in cloud water, the compounds are likely relatively stable toward hydrolysis. However, possible hydrolysis products of many of the organosulfates were also observed (Table 4). Overall, these observations are consistent with suggested acid-catalyzed organosulfate formation via the
formation of aldol condensation products, followed by reaction with sulfuric acid, during cloud droplet evaporation (Nguyen et al. 2012b).

      Biomass burning is one of the largest sources of methylglyoxal (Fu et al., 2008), and aqueous-phase reactive uptake of methylglyoxal has been suggested to be a significant source of organic aerosol (Fu et al., 2009; Zhao et al., 2006) through the formation of carboxylic acids, organosulfates, and
oligomers (Altieri et al., 2008; Ervens et al., 2011). During wildfire influence, oligomeric species were observed in cloud water samples, with the addition of up to six $C_3H_4O_2$ (methylglyoxal) monomers to glycolic acid, pyruvic acid, malonic acid, malic acid, glyoxylic acid, and succinic acid base compounds (Fig. 5). The masses corresponding to these oligomeric series were also observed by Altieri et al. (2008) in bulk aqueous-phase laboratory studies of the photooxidation of methylglyoxal, suggesting the
importance of methylglyoxal-hydroxyl radical reactions for oligomer formation. Notably, pyruvic acid is one of the most abundant photooxidation products of methylglyoxal (Altieri et al., 2006; Lim et al., 2013). Modeling suggests that methylglyoxal oligomerization primarily occurs in aqueous aerosols, rather than during cloud processing (Lim et al., 2013; Tan et al., 2010). Aqueous-phase methylglyoxal reaction products are expected to remain largely in the aerosol phase upon cloud droplet evaporation (Heaton et
al., 2009; Loeffler et al., 2006). Syringol (2,6-dimethoxyphenol) and guaiacol (2-methoxyphenol) are also emitted in significant quantities from biomass burning, and previous studies have examined the bulk aqueous-phase photooxidation of syringol and guaiacol (Yu et al., 2014, 2016), showing production of several CHO compounds that were also observed in the wildfire-influenced cloud water studied here (Table 3). Notably, aqueous SOA formation from phenolic compounds has been shown to enhance light
absorption in the UV-visible region (Yu et al., 2014), suggesting that these brown carbon compounds in the cloud water may be important upon cloud droplet evaporation.

**4 Conclusions and atmospheric implications**

This study represents only the second determination of high molecular weight organic molecular
composition in cloud water at Whiteface Mountain, NY (Sagona et al., 2014) and the first using ultrahigh-resolution mass spectrometry. The average O:C ratios of the oxygenated high mass organic compounds (negative ion mode, $m/z$ 100-1000) increased with decreasing cloud water pH, suggesting the influence of aqueous acid-catalyzed oxidation reactions contributing to the dissolved organic compounds. However, without knowledge of the organic aerosol composition prior to cloud formation, it is not possible to
distinguish between aqueous aerosol and cloud droplet processes in this study. Cloud water samples from





different air masses featured notable differences in the detected compounds. A higher absolute number and number fraction of CHNO compounds were observed during the wildfire and urban air mass periods, compared to biogenic influence, likely due to greater $NO_x$ influence. In addition, observations of likely dinitroaromatics during all air mass influences, potentially from aqueous nitration of nitroaromatic

compounds, have important implications because of their light-absorbing and mutagenic properties (Purohit and Basu, 2000; Zhang et al., 2011, 2013). During wildfire influence, the cloud water showed evidence of aqueous SOA formation, including oligomer formation involving methylglyoxal (Altieri et al., 2008; Yasmeen et al., 2010) and aqueous-phase reactions of syringol and guaiacol (Yu et al., 2014, 2016). Monoterpene-derived organosulfates and organonitrates (Surratt et al., 2008) were observed in the

cloud water during all air mass influences, similar to previous cloud water studies (Boone et al., 2015; Pratt et al., 2013). Notably, long chain alkane-derived organosulfates (Riva et al., 2016) were observed in cloud water when urban influence was present. Future work comparing cloud water composition with aqueous aerosol is needed to isolate in-cloud processes from aqueous-aerosol processes.

*Competing Interests:* The authors declare that they have no competing financial interests.

*Acknowledgements:* This study was supported by the New York State Energy Research and Development Authority (NYSERDA, contract 53968) and a University of Michigan (UM) College of Literature, Science, and the Arts Honors Grant to Z. Peng. R. Cook was supported by a UM Rackham Graduate School Merit Fellowship. Y.-H. Lin was supported by a postdoctoral fellowship from the UM College of the Literature, Science, and the Arts and the Michigan Society of Fellows. Z. Peng was supported by a

UM Department of Chemistry Summer Undergraduate Research Program Fellowship. High-resolution mass spectrometry analyses were performed at the Environmental Molecular Sciences Laboratory (EMSL), a national scientific user facility located at the Pacific Northwest National Laboratory (PNNL) and sponsored by the Office of Biological and Environmental Research of the U.S Department of Energy (DOE). PNNL is operated for DOE by Battelle Memorial Institute under Contract No. DE-AC06-76RL0

1830. ALSC cloud water collection is supported by the New York State Department of Environmental Conservation and NYSERDA. The authors thank Lynn Mazzoleni (Michigan Technological University) for discussion about the solid phase extraction methodology, as well as Stephen McNamara (University of Michigan) for research assistance. The authors gratefully acknowledge the NOAA Air Resources Laboratory for the provision of the HYSPLIT transport and dispersion model used in this work.



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





**Table 1.** Summary of the sampling times, as well as inorganic ion concentrations, pH, and TOC concentrations, for cloud water samples from Whiteface Mountain Observatory. Cloud water samples were collected with 3 h resolution (local time: EDT). Values below the method limit of detection for $Na^+$ are noted as "bdl". 95% confidence intervals are shown for Aug.-Sept. 2014 averages.

| Event | Sample ID | Sampling Start Time | $SO_4^{2-}$ µM | $NO_3^-$ µM | $Cl^-$ µM | $Ca^{2+}$ µM | $Mg^{2+}$ µM | $Na^+$ µM | $K^+$ µM | $NH_4^+$ µM | pH | TOC mg L$^{-1}$ |
|---|---|---|---|---|---|---|---|---|---|---|---|---|
| Biogenic | A1 | 8/16/2014 18:00 | 4.6 | 4.9 | 0.3 | 1.6 | 0.8 | bdl | 0.45 | 7.6 | 5.25 | 0.73 |
| | A2 | 8/17/2014 06:00 | 9.7 | 6.3 | 0.3 | 0.7 | 0.4 | bdl | 0.30 | 9.2 | 4.79 | 1.46 |
| | A3 | 8/17/2014 18:00 | 3.6 | 3.2 | 0.5 | 0.5 | 0.3 | bdl | 0.37 | 6.7 | 4.94 | 2.16 |
| Urban | B1 | 9/2/2014 06:00 | 69.4 | 30.7 | 2.7 | 2.2 | 0.8 | 4.3 | 0.87 | 79.5 | 4.18 | 2.11 |
| | B2 | 9/2/2014 18:00 | 10.1 | 10.2 | 0.7 | 0.3 | 0.2 | bdl | 0.02 | 12.7 | 4.86 | 1.19 |
| Wildfire | C1 | 8/4/2014 18:00 | 100.1 | 158.3 | 6.5 | 18.0 | 6.2 | 0.9 | 5.13 | 311.8 | 4.05 | 16.6 |
| | C2 | 8/5/2014 06:00 | 103.2 | 176.6 | 7.9 | 20.8 | 6.6 | 1.3 | 5.38 | 373.5 | 4.10 | 16.6 |
| | C3 | 8/6/2014 18:00 | 9.4 | 13.3 | 1.3 | 5.3 | 1.1 | 0.4 | 1.30 | 33.0 | 4.54 | 7.86 |
| Aug-Sept 2014 average values | | | 32 ± 9 | 38 ± 12 | 9 ± 8 | 7 ± 3 | 2 ± 1 | 11 ± 2 | 1.6 ± 0.5 | 66 ± 11 | 4.8 ± 0.1 | 3.1 ± 0.8 |



**Table 2.** Identified SOA tracers (*m/z*, neutral molecular formula, suggested VOC precursor, and compound type, based on previous laboratory chamber studies) in each cloud water sample, with observed potential hydrolysis products of likely organosulfates noted.

| $[M-H]^-$ (*m/z*) | Molecular Formula | VOC Precursors | Observed in Cloud Water Samples | Observed Potential Hydrolysis Products |
|---|---|---|---|---|
| 223.0282 | $C_7H_{12}O_6S$ | | B1, C1, C2 | $C_7H_{12}O_3$ |
| 237.0438 | $C_8H_{14}O_6S$ | | A2, B1, C1, C2, C3 | -- |
| 279.0544 | $C_{10}H_{16}O_7S$ | α-pinene | A1, A2, A3, B1, B2, C1, C2, C3 | $C_{10}H_{16}O_4$ |
| 281.0700 | $C_{10}H_{18}O_7S$ | | A1, A2, A3, B1, B2, C1, C2, C3 | $C_{10}H_{18}O_4$ |
| 297.0650 | $C_{10}H_{18}O_8S$ | | A1, A2, A3, B1, B2, C1, C2, C3 | $C_{10}H_{18}O_5$ |
| 249.0802 | $C_{10}H_{18}O_5S$ | β-pinene | A1, A2, A3, B1, B2, C1, C2, C3 | -- |
| 263.0595 | $C_{10}H_{16}O_6S$ | | A1, A2, A3, B1, B2, C1, C2, C3 | -- |
| 239.0231 | $C_7H_{12}O_7S$ | | B1, C1 | $C_7H_{12}O_4$ |
| 251.0595 | $C_9H_{16}O_6S$ | | A1, A2, B1, C1, C2, C3 | $C_9H_{16}O_3$ |
| 267.0545 | $C_9H_{16}O_7S$ | d-limonene | A2, B1, C1, C2, C3 | $C_9H_{16}O_4$ |
| 279.0544 | $C_{10}H_{16}O_7S$ | | A1, A2, A3, B1, B2, C1, C2, C3 | $C_{10}H_{16}O_4$ |
| 281.0700 | $C_{10}H_{18}O_7S$ | | A1, A2, A3, B1, B2, C1, C2, C3 | $C_{10}H_{18}O_4$ |
| 253.0388 | $C_8H_{14}O_7S$ | | B1, C1, C2 | -- |
| 283.0493 | $C_9H_{16}O_8S$ | | A1, A2, A3, B1, B2, C1, C2, C3 | $C_9H_{16}O_5$ |
| 279.0544 | $C_{10}H_{16}O_7S$ | α-terpinene | A1, A2, A3, B1, B2, C1, C2, C3 | $C_{10}H_{16}O_4$ |
| 281.0700 | $C_{10}H_{18}O_7S$ | | A1, A2, A3, B1, B2, C1, C2, C3 | $C_{10}H_{18}O_4$ |
| 283.0857 | $C_{10}H_{20}O_7S$ | | A1, A2, A3, B1, B2, C1, C2, C3 | $C_{10}H_{20}O_4$ |
| 297.0650 | $C_{10}H_{18}O_8S$ | | A1, A2, A3, B1, B2, C1, C2, C3 | $C_{10}H_{18}O_5$ |
| 296.0446 | $C_9H_{15}NO_8S$ | | B1, B2, C1 | $C_{10}H_{18}O_5$ |
| 294.0653 | $C_{10}H_{17}NO_7S$ | Monoterpenes | A1, A2, A3, B1, B2, C3 | $C_{10}H_{18}O_5S$; $C_{10}H_{17}NO_4$ |
| 342.0500 | $C_{10}H_{17}NO_{10}S$ | | A1, A2, A3, B1, B2, C1, C2, C3 | $C_{10}H_{18}O_8S$; $C_{10}H_{17}NO_7$; |
| 209.0489 | $C_7H_{14}O_5S$ | Dodecane | B1, C1 | -- |
| 267.0545 | $C_9H_{16}O_7S$ | Decalin/ Cyclodecane | A2, B1, C1, C2, C3 | -- |
| 299.0806 | $C_{10}H_{20}O_8S$ | Decalin | B1, C1, C2, C3 | $C_{10}H_{20}O_5$ |
| 311.0442 | $C_{10}H_{16}O_9S$ | Decalin | A2, B1, C1, C2, C3 | $C_{10}H_{16}O_6$ |
| 307.0857 | $C_{12}H_{20}O_7S$ | Dodecane | A1, A2, B1, B2, C1, C2, C3 | $C_{12}H_{20}O_4$ |

(The "VOC Precursors" column also spans a merged cell labeled "Monoterpene-derived organosulfates and nitroxy-organosulfates (Iinuma et al., 2007a, 2007b, 2009; Nozière et al., 2010; Surratt et al., 2008)" covering the monoterpene-related rows, and "Aliphatic organosulfates (Riva et al., 2016)" covering the aliphatic rows.)



**Table 3.** Identified SOA tracers (*m/z*, molecular formula, and suggested VOC precursor, based on bulk aqueous laboratory studies) in each cloud water sample.

| [M-H]⁻ (*m/z*) | Molecular Formula | VOC Precursors | Compound Type | Observed in Cloud Water Samples |
|---|---|---|---|---|
| 213.0769 | $C_{10}H_{14}O_5$ | α-Pinene | Aqueous monoterpene SOA tracers (Aljawhary et al., 2016) | A1, A2, A3, B1, B2, C1, C2, C3 |
| 229.0718 | $C_{10}H_{14}O_6$ | | | A1, A2, A3, B1, B2, C1, C2, C3 |
| 157.0506 | $C_7H_{10}O_4$ | Methyl vinyl ketone | Aqueous isoprene SOA tracers (Renard et al., 2015) | B1, C1, C2, C3 |
| 159.0663 | $C_7H_{12}O_4$ | | | B1, C3 |
| 199.0612 | $C_9H_{12}O_5$ | | | A2, A3, B1, B2, C1, C2, C3 |
| 320.7606 | $C_{13}H_{22}O_9$ | Isoprene | Aqueous isoprene SOA tracers (Nguyen et al., 2012a) | A2, B1, C1, C2, C3 |
| 318.7835 | $C_{14}H_{24}O_8$ | | | A1, A2, A3, B1, B2, C1, C2, C3 |
| 330.7337 | $C_{14}H_{20}O_9$ | | | A1, A3, B1, B2, C1, C2, C3 |
| 348.7242 | $C_{14}H_{22}O_{10}$ | | | A3, B1, C1, C2, C3 |
| 364.7012 | $C_{14}H_{22}O_{11}$ | | | C1, C2 |
| 346.7471 | $C_{15}H_{24}O_9$ | | | A1, A2, A3, B1, B2, C1, C2, C3 |
| 392.6649 | $C_{15}H_{22}O_{12}$ | | | C1, C2 |
| 406.6650 | $C_{16}H_{24}O_{12}$ | | | C1, C2, C3 |
| 436.6420 | $C_{17}H_{26}O_{13}$ | | | C1, C2 |
| 185.0455 | $C_8H_{10}O_5$ | Syringol (2,6-dimethoxyphenol) | Aqueous biomass burning SOA tracers (Yu et al., 2014, 2016) | B1, C1, C2, C3 |
| 251.0562 | $C_{12}H_{12}O_6$ | | | B1, C1, C2, C3 |
| 267.0511 | $C_{12}H_{12}O_7$ | | | B1, C2, C3 |
| 289.0718 | $C_{15}H_{14}O_6$ | | | B1, C1, C2, C3 |
| 291.0874 | $C_{15}H_{16}O_6$ | | | B1, B2, C1, C2, C3 |
| 309.0980 | $C_{15}H_{18}O_7$ | | | A1, A2, A3, B1, B2, C1, C2, C3 |
| 339.0722 | $C_{15}H_{16}O_9$ | | | B1, C1, C2, C3 |
| 305.1030 | $C_{16}H_{18}O_6$ | | | A2, B1, B2, C1, C2, C3 |
| 262.7625 | $C_{13}H_{12}O_6$ | Guaiacol (2-methoxyphenol) | | B1, C1, C2, C3 |
| 260.7854 | $C_{14}H_{14}O_5$ | | | B1, C1, C2, C3 |
| 274.7490 | $C_{14}H_{12}O_6$ | | | B1, C3 |
| 276.7624 | $C_{14}H_{14}O_6$ | | | A2, B1, B2, C1, C2, C3 |



**Table 4.** Identity of possible organosulfates, previously observed in the laboratory from the evaporation and re-dissolution of an aqueous solution of d-limonene SOA and sulfuric acid (Nguyen et al., 2012b), in each cloud water sample, with observed potential hydrolysis products noted.

| [M-H]$^-$ ($m/z$) | Molecular Formula | Observed in Cloud Water Samples | Observed Potential Hydrolysis Products |
|---|---|---|---|
| 223.0280 | $C_7H_{12}O_6S$ | B1, C1, C2 | $C_7H_{12}O_3$ |
| 225.0437 | $C_7H_{14}O_6S$ | B1, C1, C2 | -- |
| 239.0229 | $C_7H_{12}O_7S$ | B1, C1 | $C_7H_{12}O_4$ |
| 235.0280 | $C_8H_{12}O_6S$ | B1, C1 | -- |
| 237.0436 | $C_8H_{14}O_6S$ | B1, C1, C2, C3 | -- |
| 251.0229 | $C_8H_{12}O_7S$ | B1, C1, C2 | -- |
| 253.0386 | $C_8H_{14}O_7S$ | B1, C1, C2 | -- |
| 267.0179 | $C_8H_{12}O_8S$ | B1, C1, C2 | $C_8H_{12}O_5$ |
| 249.0438 | $C_9H_{14}O_6S$ | B1, C1, C2, C3 | -- |
| 251.0594 | $C_9H_{16}O_6S$ | B1, C1, C2, C3 | -- |
| 263.0231 | $C_9H_{12}O_7S$ | B1 | -- |
| 265.0387 | $C_9H_{14}O_7S$ | B1, B2, C1, C2, C3 | -- |
| 267.0543 | $C_9H_{16}O_7S$ | B1, C1, C2, C3 | -- |
| 281.0335 | $C_9H_{14}O_8S$ | B1, C1, C2, C3 | $C_9H_{14}O_5$ |
| 297.0285 | $C_9H_{14}O_9S$ | B1, C1, C2 | $C_9H_{14}O_6$ |
| 299.0441 | $C_9H_{16}O_9S$ | C1 | $C_9H_{16}O_6$ |
| 263.0593 | $C_{10}H_{16}O_6S$ | B1, B2, C1, C2, C3 | -- |
| 277.0387 | $C_{10}H_{14}O_7S$ | B1, C1, C2, C3 | -- |
| 279.0542 | $C_{10}H_{16}O_7S$ | B1, B2, C1, C2, C3 | $C_{10}H_{16}O_4$ |
| 281.0698 | $C_{10}H_{18}O_7S$ | B1, B2, C1, C2, C3 | $C_{10}H_{18}O_4$ |
| 293.0337 | $C_{10}H_{14}O_8S$ | B1, C1, C2 | $C_{10}H_{14}O_5$ |
| 295.0493 | $C_{10}H_{16}O_8S$ | B1, B2, C1, C2, C3 | $C_{10}H_{16}O_5$ |
| 297.0648 | $C_{10}H_{18}O_8S$ | B1, B2, C1, C2, C3 | $C_{10}H_{18}O_5$ |
| 311.0441 | $C_{10}H_{16}O_9S$ | B1, C1, C2, C3 | $C_{10}H_{16}O_6$ |
| 327.0388 | $C_{10}H_{16}O_{10}S$ | B1, C1, C2 | $C_{10}H_{16}O_7$ |
| 329.0546 | $C_{10}H_{18}O_{10}S$ | C1 | $C_{10}H_{18}O_7$ |
| 423.0961 | $C_{16}H_{24}O_{11}S$ | B1, C1, C2 | $C_{16}H_{24}O_8$ |
| 441.1072 | $C_{16}H_{26}O_{12}S$ | B1, C1, C2 | $C_{16}H_{26}O_9$ |
| 457.1030 | $C_{16}H_{26}O_{13}S$ | B1, C1, C2 | $C_{16}H_{26}O_{10}$ |



| [M-H]⁻ ($m/z$) | Molecular Formula | Observed in Cloud Water Samples | Observed Potential Hydrolysis Products |
|---|---|---|---|
| 439.1279 | $C_{17}H_{28}O_{11}S$ | B1, C1, C2, C3 | $C_{17}H_{28}O_8$ |
| 455.1234 | $C_{17}H_{28}O_{12}S$ | B1, C1, C2, C3 | $C_{17}H_{28}O_9$ |
| 469.1028 | $C_{17}H_{26}O_{13}S$ | C1, C2 | -- |
| 471.1177 | $C_{17}H_{28}O_{13}S$ | C1, C2 | $C_{17}H_{28}O_{10}$ |
| 451.1275 | $C_{18}H_{28}O_{11}S$ | B1, B2, C1, C2, C3 | $C_{18}H_{28}O_8$ |
| 465.1075 | $C_{18}H_{26}O_{12}S$ | C1, C2 | -- |
| 467.1230 | $C_{18}H_{28}O_{12}S$ | C1, C2 | $C_{18}H_{28}O_9$ |
| 469.1385 | $C_{18}H_{30}O_{12}S$ | B1, C1, C2 | $C_{18}H_{30}O_9$ |
| 483.1180 | $C_{18}H_{28}O_{13}S$ | B1, C1, C2 | -- |
| 485.1330 | $C_{18}H_{30}O_{13}S$ | B1, C1, C2 | $C_{18}H_{30}O_{10}$ |
| 499.1127 | $C_{18}H_{28}O_{14}S$ | C1 | -- |
| 481.1385 | $C_{19}H_{30}O_{12}S$ | B1, C1, C2, C3 | $C_{19}H_{30}O_9$ |
| 483.1537 | $C_{19}H_{32}O_{12}S$ | B1, C1, C2, C3 | $C_{19}H_{32}O_9$ |
| 497.1341 | $C_{19}H_{30}O_{13}S$ | C1 | -- |
| 513.1276 | $C_{19}H_{30}O_{14}S$ | C1, C2 | -- |
| 513.1653 | $C_{20}H_{34}O_{13}S$ | B1, C3 | $C_{20}H_{34}O_{10}$ |




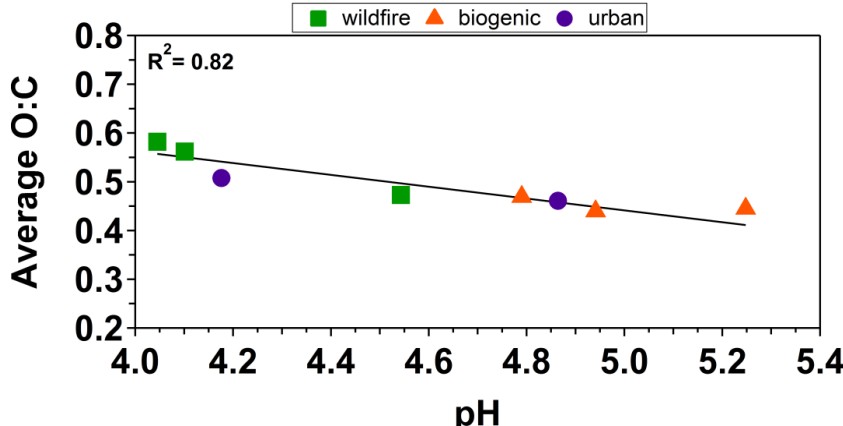

**Figure 1.** Correlation between cloud water pH and average O:C ratio (negative ion mode, *m/z* 100-1000)

by air mass source influence.





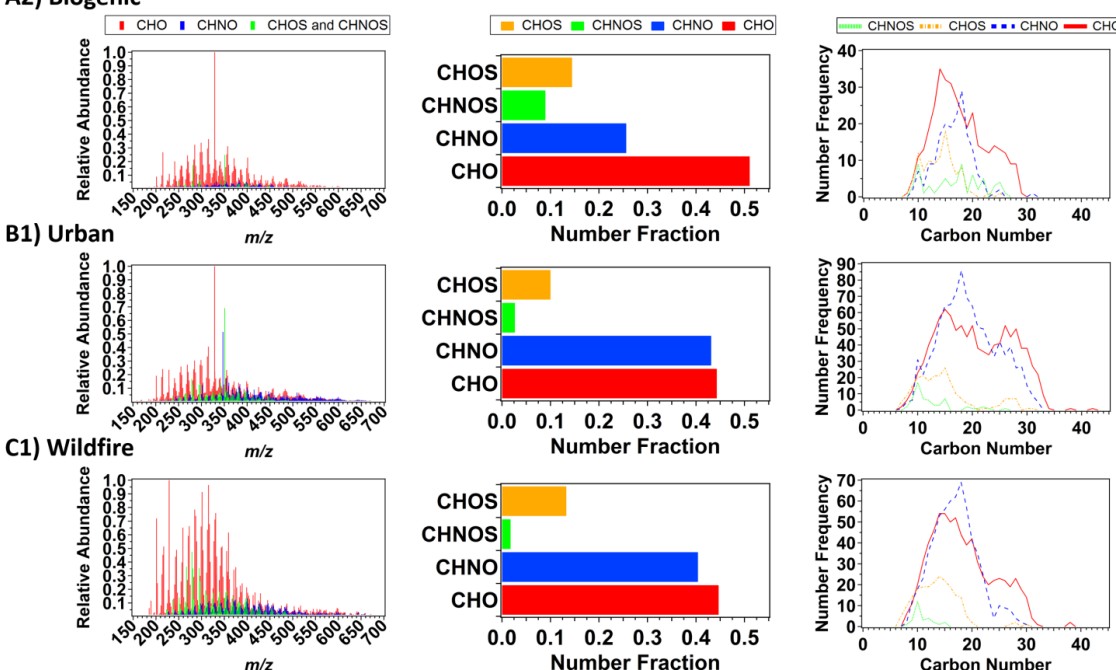

**Figure 2.** (-)ESI-FTICR-MS (left), compound type number fractions (middle), and carbon number frequency distribution (right) of compound types (CHO, CHNO, CHOS, and CHNOS) for representative cloud water samples collected on (a) August 17, 2014 06:00-09:00 EDT (A2), under northern forest (biogenic) influence combined with urban pollution, (b) September 2, 2014 06:00-09:00 EDT (B1), under urban pollution influence, and (c) August 4, 2014 18:00-21:00 EDT (C1), under the influence of Canadian wildfires. Mass spectra and carbon number distribution for the remaining samples are shown in Figure S2 and S3, respectively.

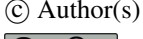



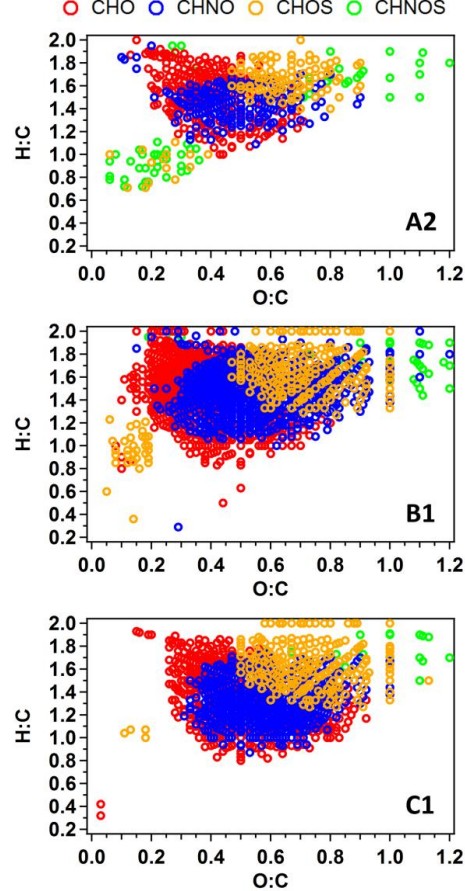

**Figure 3.** Van Krevelen diagrams as a function of compound type (CHO, CHNO, CHOS, and CHNOS)

for the representative cloud water samples collected on (A2) August 17, 2014 06:00-09:00 EDT, (B1)

September 2, 2014 06:00-09:00 EDT, and (C1) August 4, 2014 18:00-21:00 EDT. Van Krevelen diagrams

5     for the remaining samples are shown in Figure S4.





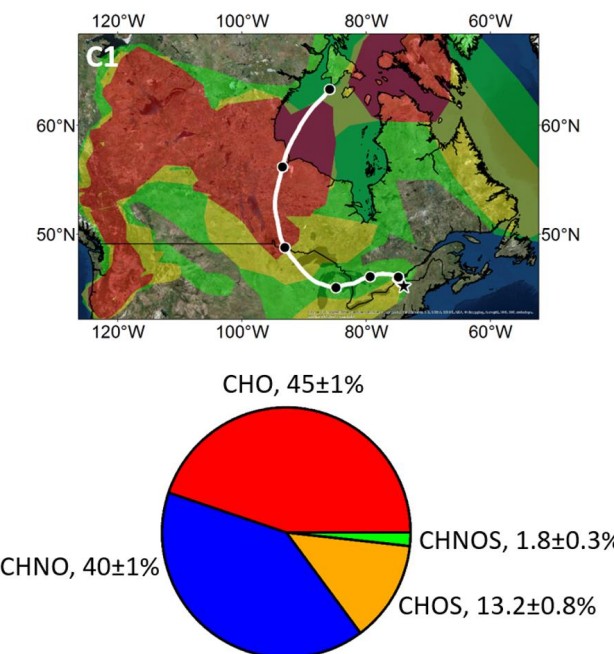

**Figure 4.** NOAA HYSPLIT 144 h (6 day) backward air mass trajectory (white line, with 6 h resolution, shown as black dots), and corresponding changes in cloud water molecular composition, shown as compound type number fractions during the wildfire event (sample C1). Green, yellow, and red shading on the map represents low, moderate, and high amounts of smoke, respectively, as defined by the NOAA Hazard Mapping System Fire and Smoke Product (http://www.ospo.noaa.gov/Products/land/hms.html).





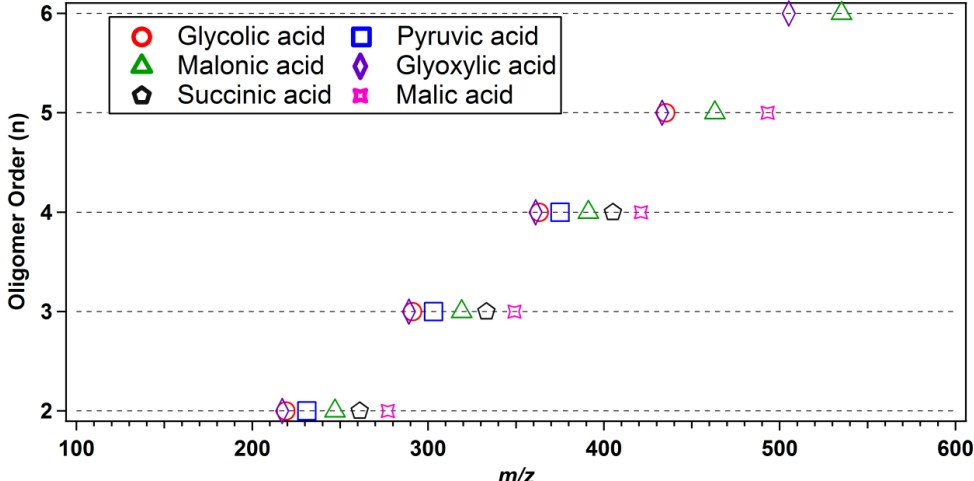

**Figure 5.** Order of oligomer (n) corresponding to the number of methylglyoxal ($C_3H_4O_2$) additions ($\Delta m/z$=72.02113) to specific parent organic acids (listed in the legend), proposed to form through aqueous processing and measured in cloud water during the wildfire-influenced period (C1-C3). Compound identification was based on comparison of elemental formulas calculated from (-)ESI-FTICR-MS data with the oligomeric series previously reported by Altieri et al. (2008) from the aqueous reactions of methylglyoxal + ·OH.