# Peer review of "Biogenic, urban, and wildfire influences on the molecular composition of dissolved organic compounds in cloud water"

_Atmospheric Chemistry and Physics, 2017_

## Referee Comment (RC1) · Anonymous Referee #1 · 25 Aug 2017

General comments This study reports the molecular composition of cloud water collected under urban, biogenic, and biomass burning influences revealing key differences among dissolved organic compounds in each condition. The results are clearly presented, concise, and contain sufficient detail. I have only minor suggestions for strengthening this work. Upon revision, I would recommend that the editor accept this manuscript.

Specific comments

Abstract: The abstract would benefit from a concluding sentence in the abstract that draws the reader's attention to the most significant findings of this work, or to the larger

takeaway from the measurements.

Pg 4 Line 4: What were TOC concentrations before the concentration step? Since there was a concentration step, it's not clear what giving these values tells the reader, except that the concentrated samples were probably above LOD for most techniques. Even an estimate of the increase (doubled, 10-fold) would be useful here. Line 11: Why negative ion mode? I'd think that you would miss any ammonium-based oligomers including imines. Negative ion mode captures acids well, but not reduced nitrogen compounds. Could the authors at least comment on this point here?

Pg 8 Lines 16-17: I'm not sure the data support the claim that acidity (as opposed to BB-specific VOCs, their concentration, or pyrolysis itself) is responsible for the increase in oligomeric material. It's certainly possible, but it's also possible that the BB cloud water simply had lower pH AND more oligomeric compounds. Higher concentrations of the same compounds could facilitate increased oligomerization too, right? Please revise to avoid overstating the connection.

Technical corrections Pg 1 Line 26: "with a focus" Line 31: "was positively correlated"

Pg 2 Line 4: "water soluble organic gases" so as to distinguish $SO_2$ from the organic compounds alluded to here Line 6: "depending on their solubility" for concision

Pg 4 Line 5: "including some organosulfate" right? Not all? I believe organosulfates are reported here.

Pg 5 Line 6: "compose" not comprise

Pg 9 Line 12: "succinic acid based"

―――――――――――――――

---

## Author Comment (AC1) · 26 Sep 2017

**acp-2017-628: Response to Anonymous Referee #1**

**We would like to thank the referee for their comments and suggestions. Below, the original comment provided by the referee is shown in gray. Our response, with revisions noted, are in bold font.**

General comments
This study reports the molecular composition of cloud water collected under urban, biogenic, and biomass burning influences revealing key differences among dissolved organic compounds in each condition. The results are clearly presented, concise, and contain sufficient detail. I have only minor suggestions for strengthening this work. Upon revision, I would recommend that the editor accept this manuscript.

Specific comments

Abstract: The abstract would benefit from a concluding sentence in the abstract that draws the reader's attention to the most significant findings of this work, or to the larger takeaway from the measurements.

> **The following sentence was added to the end of the abstract: "Overall, cloud water molecular composition depended on air mass source influence and reflected aqueous-phase reactions involving biogenic, urban, and biomass burning precursors."**

Pg 4 Line 4: What were TOC concentrations before the concentration step? Since there was a concentration step, it's not clear what giving these values tells the reader, except that the concentrated samples were probably above LOD for most techniques. Even an estimate of the increase (doubled, 10-fold) would be useful here.

> **The cloud water TOC concentrations prior to the concentration step are shown in Table 1 of the main text, as now noted in the methods section, as requested.**

Line 11: Why negative ion mode? I'd think that you would miss any ammonium-based oligomers
including imines. Negative ion mode captures acids well, but not reduced nitrogen compounds. Could the authors at least comment on this point here?

> **Negative ion mode was used to target CHO compounds, which are more easily ionized in negative ion mode and are the primary component of cloud water. We now note on P4 L15 that we are targeting oxidized organic compounds by creating negative ions. While examining both positive and negative ion mode would have been useful, limited access to instrumentation required only one mode to be chosen for these samples.**

Pg 8 Lines 16-17: I'm not sure the data support the claim that acidity (as opposed to BB-specific VOCs, their concentration, or pyrolysis itself) is responsible for the increase in oligomeric material. It's certainly possible, but it's also possible that the BB cloud water simply had lower pH AND more oligomeric compounds. Higher concentrations of the same compounds could facilitate increased oligomerization too, right? Please revise to avoid overstating the connection.

**This is an excellent point. We have revised this sentence (moved to P8 L 30-34) as follows: "Notably, there appear to be a greater diversity of oligomeric compounds present in the wildfire samples (Fig. 2, Fig. S2), which is likely a combination of the identities of the specific organic compounds present at high concentrations in the smoke, as well as the potential role of acidity (as observed through cloud water acidity), resulting in the production of the observed oligomeric species."**

Technical corrections Pg 1 Line 26: "with a focus" Line 31: "was positively correlated"

**The suggested changes were made.**

Pg 2 Line 4: "water soluble organic gases" so as to distinguish SO2 from the organic compounds alluded to here Line 6: "depending on their solubility" for concision

**The suggested changes were made.**

Pg 4 Line 5: "including some organosulfate" right? Not all? I believe organosulfates are reported here.

**This is correct. The suggested change was made.**

Pg 5 Line 6: "compose" not comprise

**The suggested change was made.**

Pg 9 Line 12: "succinic acid based"

**The suggested change was made.**

---

## Referee Report (RR1)

**Review 2 on the manuscript « Biogenic, urban, and wildfire influences on the molecular composition of dissolved organic compounds in cloud water" by Cook et al.**

My main comments have been addressed in the author's reply to Reviewer 1 and in the revised version of the manuscript. I only have a few more comments/questions which, depending on the answers, might require some additional small changes in the manuscript.

**1. Acid-catalyzed reactions vs Fenton processes (radical generation)**

As for Reviewer 1, my main concern was the conclusion that the correlation between O/C ratios and pH necessarily meant the occurrence of acid-catalyzed reactions. But such reactions are extremely slow over the range of pH of 4-5 reported here (typically << 10-7 s-1, thus take weeks or months). By contrast, radical production in aqueous media, which is controlled by Fenton processes, is strongly pH-dependent and a much more plausible explanation for these results. I saw that the authors have taken this comment into account by adding "and radical processes" next to "acid-catalyzed". I would suggest writing "or" rather than "and", or any other formulation pointing more strongly towards radical processes.

**2. Acid-catalyzed reactions vs biological processes**

Another explanation that was not considered is that the oligomeric material found in the cloudwater could be of microbial origin. There is now an extensive literature reporting the presence of microorganisms in cloud water, alive and in full capacity to metabolize (process) organic compounds at rates that are in competition with non-biological reactions (see for instance Vaitilingom et al., ACP 11, 8721, 2011). Microorganisms are also well-known to produce organic acids, which would affect the pH of the medium. Thus, the observed correlation O/C vs pH could be entirely biologically-driven. The presence of microorganisms and of their metabolites in cloud water might not be easy to link to specific sources as they are present in all kinds of environments.

Could the authors discuss to which extend this could be the case (or ruled out) ?

**3. Potential formation of oligomers in the electrospray**

The samples do not seem to be ran through a column before to be introduced into the electrospray of the spectrometer. Such direct injection (or infusion) has been known in the past to produce spurious oligomers (= not present in the initial samples) in the ionizing source. Could the author discuss the probability of this happening in their analyses ?

**4. Aerosols: source or target of aqueous-phase processes ?**

This comment addresses the argument given in introduction to justify the relevance of this study. I am not asking the authors to necessarily change their introduction at this point, but at least to keep this comment in mind in future works.

Indeed, the idea that aqueous-phase processes can affect aerosol composition and SOA formation has been circulating for some years. This is, however, difficult to justify as cloud (or fog) droplets are in very small numbers (a few 100 cm-3 at the most) compared to the number of aerosol particles in most regions (1000 to 100 000 cm-3). Thus only one in 10 to one in 1000 aerosol particle might be affected by aqueous-phase processes. This is far from affecting the composition of the entire aerosol population (for instance, it is hard to imagine any significant effects on the overall optical property), let alone SOA formation, which produces many 1000's particles cm-3.

Thus, a better argument for the study of the organic composition of cloudwater could be to explain the formation of specific compounds, that could be used as tracers for aqueous processes for instance, or that have specific properties.

---

## Author Response (AR2)

**acp-2017-628: Response to Reviewer 2**

**We thank the reviewer for their comments and suggestions. Below, the original comment provided by the reviewer is shown in gray. Our response, with revisions noted, is in bold font.**

My main comments have been addressed in the author's reply to Reviewer 1 and in the revised version of the manuscript. I only have a few more comments/questions which, depending on the answers, might require some additional small changes in the manuscript.

**1. Acid-catalyzed reactions vs Fenton processes (radical generation)**
As for Reviewer 1, my main concern was the conclusion that the correlation between O/C ratios and pH necessarily meant the occurrence of acid-catalyzed reactions. But such reactions are extremely slow over the range of pH of 4-5 reported here (typically << 10-7 s-1, thus take weeks or months). By contrast, radical production in aqueous media, which is controlled by Fenton processes, is strongly pH-dependent and a much more plausible explanation for these results. I saw that the authors have taken this comment into account by adding "and radical processes" next to "acid-catalyzed". I would suggest writing "or" rather than "and", or any other formulation pointing more strongly towards radical processes.

> **As now clarified on Page 3 Lines 9-14, "the molecular composition of the cloud water reflects both the precursor aerosol and any aqueous-phase processing that occurred within the cloud droplets, as well as within the aqueous aerosol prior to cloud droplet activation and following cloud droplet evaporation." As described in the review by Ervens et al. (2011) and now clarified on Page 3 Lines 2-3, the formation of high molecular weight compounds is expected to be favored within aqueous aerosol compared to cloud water due to higher rates of the self-reactions that form oligomers. pH increases with dilution, so while the cloudwater pH was ~4-5, the precursor aerosol would have a much lower pH, making acid-catalyzed reactions within the precursor aerosol potentially responsible for the observed oxidized compounds. To clarify this, we reworded the sentence on Page 1 Lines 29-32 to state: "Cloud water acidity was positively correlated with the average oxygen:carbon ratio of the organic constituents, suggesting the possibility for aqueous acid-catalyzed (prior to cloud droplet activation, or during/after cloud droplet evaporation) and/or radical (within cloud droplet) oxidation processes." The sentence on Page 10 Lines 9-12 was also reworded similarly.**

**2. Acid-catalyzed reactions vs biological processes**
Another explanation that was not considered is that the oligomeric material found in the cloudwater could be of microbial origin. There is now an extensive literature reporting the presence of microorganisms in cloud water, alive and in full capacity to metabolize (process) organic compounds at rates that are in competition with non-biological reactions (see for instance Vaitilingom et al., ACP 11, 8721, 2011). Microorganisms are also well-known to produce organic acids, which would affect the pH of the medium. Thus, the observed correlation O/C vs pH could be entirely biologically-driven. The presence of microorganisms and of their metabolites in cloud water might not be easy to link to specific sources as they are present in all

kinds of environments. Could the authors discuss to which extend this could be the case (or ruled out)?

>**It is possible that microorganisms could contribute to aqueous processing of high molecular weight compounds in cloud water; however, we are not aware of any published literature that shows high mass oligomer formation in cloud water via microorganisms. Vaïtilingom et al. (2011) examined cloud water biodegradation of carboxylic acids with molecular masses of 118 Da and lower, which could not be determined in our study, which measured compounds above *m/z* 150. Without microbial measurements, we cannot comment with respect to the observed differences between the various air masses (e.g. lower pH and higher O/C for wildfire air masses, higher pH and lower O/C for biogenic air masses). The last 20 years of work at Whiteface Mountain, summarized by Schwab et al. (2016), suggests that increasing cloud water pH at the site is associated with reductions in $SO_2$ and $NO_x$ emissions over this time period (now stated on Page 5 Lines 24-25), suggesting that cloud water pH at the site is driven by these contributions.**

**3. Potential formation of oligomers in the electrospray**

The samples do not seem to be ran through a column before to be introduced into the electrospray of the spectrometer. Such direct injection (or infusion) has been known in the past to produce spurious oligomers (= not present in the initial samples) in the ionizing source. Could the author discuss the probability of this happening in their analyses?

>**The chosen instrument parameters were optimized during previous direct injection dissolved organic matter (DOM) characterization experiments (Kujawinski and Behn, 2006; Minor et al., 2012; Tfaily et al., 2015), as stated on Page 4, Lines 16-18. During a previous cloud water study, we conducted experiments to specifically test for organosulfate formation in the ESI source, and this was not observed (Pratt et al., 2013).**

**4. Aerosols: source or target of aqueous-phase processes?**

This comment addresses the argument given in introduction to justify the relevance of this study. I am not asking the authors to necessarily change their introduction at this point, but at least to keep this comment in mind in future works.

Indeed, the idea that aqueous-phase processes can affect aerosol composition and SOA formation has been circulating for some years. This is, however, difficult to justify as cloud (or fog) droplets are in very small numbers (a few 100 cm-3 at the most) compared to the number of aerosol particles in most regions (1000 to 100 000 cm-3). Thus only one in 10 to one in 1000 aerosol particle might be affected by aqueous-phase processes. This is far from affecting the composition of the entire aerosol population (for instance, it is hard to imagine any significant effects on the overall optical property), let alone SOA formation, which produces many 1000's particles cm-3.

Thus, a better argument for the study of the organic composition of cloudwater could be to explain the formation of specific compounds, that could be used as tracers for aqueous processes for instance, or that have specific properties.

As noted in the response to Comment #1 and now clarified on Page 3 Lines 9-12, it is not possible to separate cloud-phase processes from aqueous aerosol processes prior to cloud droplet activation by measuring only cloud water. Therefore, for many of the same reasons that the reviewer points out above, we present comparisons of these field samples with laboratory aqueous-phase studies, with specific aqueous-phase tracer compounds presented in Tables 3-4 and Figure 5 and accompanying discussion. This is now stated more clearly on Page 3 Lines 21-24: "
[revised manuscript text omitted]